# Primary Liver Transplantation vs. Transplant after Kasai Portoenterostomy for Infants with Biliary Atresia

**DOI:** 10.3390/jcm11113012

**Published:** 2022-05-26

**Authors:** Caroline P. Lemoine, John P. LeShock, Katherine A. Brandt, Riccardo Superina

**Affiliations:** Division of Transplant and Advanced Hepatobiliary Surgery, Ann & Robert H. Lurie Children’s Hospital of Chicago, Northwestern University Feinberg School of Medicine, 225 E. Chicago Avenue Box 57, Chicago, IL 60611, USA; clemoine@luriechildrens.org (C.P.L.); john.leshock@northwestern.edu (J.P.L.); kabrandt@luriechildrens.org (K.A.B.)

**Keywords:** biliary atresia, kasai portoenterostomy, primary liver transplantation, outcomes

## Abstract

Introduction: Primary liver transplants (pLT) in patients with biliary atresia (BA) are infrequent, since most babies with BA undergo a prior Kasai portoenterostomy (KPE). This study compared transplant outcomes in children with BA with or without a prior KPE. We hypothesized that pLT have less morbidity and better outcomes compared to those done after a failed KPE. Methods: A retrospective review of patients with BA transplanted at our institution was performed. Patients were included if they received a pLT or if they were transplanted less than 2 years from KPE. Outcomes were compared between those groups. Comparisons were also made based on era (early: 1997–2008 vs. modern: 2009–2020). *p* < 0.05 was considered significant. Results: Patients who received a pLT were older at diagnosis (141.5 ± 46.0 vs. KPE 67.1 ± 25.5 days, *p* < 0.001). The time between diagnosis and listing for transplant was shorter in the pLT group (44.5 ± 44.7 vs. KPE 140.8 ± 102.8 days, *p* < 0.001). In the modern era, the calculated PELD score for the pLT was significantly higher (23 ± 8 vs. KPE 16 ± 8, *p* = 0.022). Two waitlist deaths occurred in the KPE group (none in pLT, *p* = 0.14). Both the duration of transplant surgery and transfusion requirements were similar in both groups. There was a significant improvement in graft survival in transplants after KPE between eras (early era 84.3% vs. modern era 97.8%, *p* = 0.025). The 1-year patient and graft survival after pLT was 100%. Conclusions: Patient and graft survival after pLT are comparable to transplants after a failed KPE but pLT avoids a prior intervention. There was no significant difference in pre- or peri-transplant morbidity between groups other than wait list mortality. A multicenter collaboration with more patients may help demonstrate the potential benefits of pLT in patients predicted to have early failure of KPE.

## 1. Introduction

Biliary atresia (BA) is a disease characterized by inflammation and obstruction of the biliary tree leading to the development of biliary cirrhosis in infancy if left untreated [1]. It was originally deemed “uncorrectable” until Kasai described a portoenterostomy, allowing bile drainage from the liver [2]. The Kasai portoenterostomy (KPE) remains the conventionally accepted treatment of BA by most pediatric surgeons. 

The success of KPE is defined by two indices: clearance of jaundice and transplant free survival. In a recent analysis of North American results among pediatric liver centers of excellence, 49.6% of children undergoing KPE achieved a normal bilirubin post-op within 3 months of surgery, and almost 50% of all children had been transplanted or died by two years after the Kasai. Even the successful clearance of jaundice does not ensure avoidance of early liver transplant [3]. Despite this, the standard of care remains to perform a KPE in all patients with followed by a liver transplantation when it fails [4,5]. In the United States, only patients with advanced liver disease at diagnosis are candidates for a primary liver transplantation (pLT) [6]. pLT is rarely performed but has been associated with excellent survival [7,8]. 

pLT is not recommended for all patients with BA since a third to a half of the patients may avoid a liver transplant in childhood after a successful KPE [9]. However, it is currently difficult to predict which patients will develop early failure after KPE (<1–2 years post-operatively) and who could, therefore, benefit from pLT. For patients who develop early failure, a pLT may lead to superior outcomes by decreasing the waitlist morbidity and possibly mortality and reducing post-transplant complications [10,11,12]. 

Here, we present our institutional experience with pLT for patients with BA and compared them to patients who received a liver transplant early (<2 years) after a failed KPE. We hypothesized that pLT leads to superior post-transplant outcomes to transplant after KPE and is associated with a lower waitlist morbidity and mortality. 

## 2. Material and Methods 

### 2.1. Patient Selection, Definitions and Data Collection

Data were collected retrospectively from our BA and liver transplant databases. Only patients who suffered from early failure after KPE or who received a pLT at our institution between August 1997 and June 2020 were included for this comparative study. Early failure was defined as BA patients who received a liver transplant less than 2 years after KPE. Patient selection is illustrated in Figure 1. 

This retrospective study comparing patients with BA who were transplanted either after KPE or with a pLT was approved by our Institutional Review Board (IRB 2013-15357 and 2007-12989). Ultimately, a total of 99 patients were included in the KPE group and 14 patients received a pLT. 

Primary liver transplant was done in patients who were diagnosed with BA by biopsy and operative cholangiogram who had signs of advanced liver disease at presentation: portal hypertension defined by the presence of hypersplenism (thrombocytopenia, splenomegaly) or history of variceal bleed, ascites, growth failure or synthetic liver dysfunction (INR ≥ 1.7, Albumin ≤ 3.2 g/dL). 

Data collected included: patient’s characteristics (sex, prematurity); age and weight at BA diagnosis (for KPE group: age at KPE; for pLT group: age at liver biopsy showing features of BA); transplant waitlist-related data (age at listing, waitlist duration, hospital admissions while listed, number of days admitted, indication for hospital admission, cost of admissions—see below); transplant-related data (age and weight at transplant, natural Pediatric End-stage Liver Disease (PELD) score a transplant for patients listed for transplant after 2002, length of transplant surgery, intraoperative packed red blood cell transfusion requirements, surgical complications, length of post-operative mechanical ventilation, length of intensive care and hospital stay). Patient and graft status at last follow-up as well as retransplantation were also collected. 

For the purposes of evaluating the impact of practice changes over time on outcomes, we divided the experience into an early (August 1997–December 2008) and modern (January 2009—June 2020) era. Both the KPE and transplants were done primarily by a single surgeon (RS) in both eras.

Post-operative management did not change substantially from one era to the other for the post-operative KPE care. Intravenous ampicillin and gentamicin were used in all KPE patients post-operatively for 5 days followed by trimethoprim/sulfamethoxazole oral prophylaxis for 6 months. Post-operative steroids were not used routinely in either of the two eras. All KPE were done open and not laparoscopically.

Listing criteria for transplantation in the KPE group included failure to thrive despite optimal nutritional management, recurrent spontaneous bacterial peritonitis despite optimal management of ascites and hepatic synthetic failure as exemplified by vitamin K resistant INR of greater than 1.7 and albumin less the 3.0 g/100 mL.

Children who achieved at least a 2-year transplant-free survival or who achieved a serum direct bilirubin of <0.2 mg/dL after KPE were not included in the study, since these children met criteria for a successful KPE and were older and bigger than the control group.

No child survived more than two years after a failed KPE without a transplant.

Cost data were obtained through our institution’s billing department for patients transplanted at our institution after 2009. In order to ensure the data would be comparable, all cost data were converted into an inflation-adjusted measure for a chosen baseline time period (chosen as the Consumer Price Index (CPI) for 2020). The CPI medical services index (a measure of change over time in the prices of medical services) was utilized to perform this conversion (https://fred.stlouisfed.org/series/CPIMEDSL (accessed on 26 June 2021)). Instead of performing a calculation on a monthly basis, an average of the CPI values for all months in a given year was obtained and then used for calculation using the following formula: Equivalent cost in baseline period = (Cost amount) × ((CPI for baseline time period)/(CPI for time period of the charge)). The adjusted cost of hospital admissions while on the transplant waitlist included the cost of the KPE admission for patients in the KPE group. The cost data are presented in United States dollars (USD). 

### 2.2. Outcomes 

Primary outcomes included 1-year and 3-year post-transplant patient and graft survival. Secondary outcomes focused on waitlist morbidity (number of hospital admissions, days admitted while on the wait list, indication for hospital admission and waitlist duration). Additionally, the morbidity at the time of transplant was evaluated, focusing on length of surgery, intra-operative transfusion requirements and post-transplant ICU and hospital length of stay. 

### 2.3. Statistical Analysis 

Comparisons were made between the KPE and pLT groups using the independent t-test for continuous variables and the Chi-square test for categorical variables. Additionally, comparisons were made based on the era of management to account for changes and improvement in the management of patients with BA. The same statistical analyses were performed. 

The Mann–Whitney U test was used to compare billing data given its non-parametric distribution. Kaplan–Meier survival curves were obtained to compare patient and graft survival. Statistics were performed using the IBM SPSS Statistics program (version 24.0.0.0). A *p* < 0.05 was considered statistically significant. 

## 3. Results 

### 3.1. Pre-Transplant Data: Comparison of Patients Transplanted after pLT or KPE

The incidence of pLT was 12.4% (14/113). Patient’s characteristics and pre-transplant waitlist data are presented in Table 1. Patients in the pLT group were significantly older at the time of BA diagnosis (KPE 67.1 ± 25.5 vs. pLT 141.5 ± 46.0 days, *p* < 0.001). Although the time between diagnosis and listing for transplant was shorter in the pLT group (KPE 140.8 ± 102.8 vs. pLT 44.5 ± 44.7 days, *p* = 0.001), the time that was spent on the waitlist was not statistically shorter (*p* = 0.6). Neither the number of hospital admissions nor the total number of days admitted while waiting for transplant were different when comparing groups. Although there was a trend in a lower cumulative adjusted cost of hospital admissions for the pLT group, this difference failed to reach statistical significance (KPE $425,090.00 (285,282.83, 566,405.40) vs. pLT $253,004.10 (95,640.49, 431,530.70), *p* = 0.07). The reasons for and number of hospital admissions were similar in both groups, except for cholangitis. Patients from the KPE group were often admitted for cholangitis, a complication that did not occur in any infant from the pLT group (KPE 31/99, 31.3% vs. pLT 0/14, *p* = 0.014). Two deaths on the waitlist occurred in the KPE group. Although there were none in the pLT group, this difference was not significant (*p* = 0.59). 

### 3.2. Transplant and Survival Data: Comparison between pLT and KPE Groups 

Patients who received a pLT were nearly 4 weeks younger at the time of transplant (pLT 287.0 ± 82.7 days versus KPE 311.4 ± 144.0, *p* = 0.54) (Table 2). The groups were comparable in terms of severity of disease at transplant (similar growth failure based on weight z-scores and calculated PELD score). From a surgical standpoint, there was no difference in length of surgery or intraoperative packed red blood cell transfusion requirements. The number of combined returns to the operating room for any surgical complication (bleeding, thrombosis, bowel perforation or biliary complications) or procedures performed in interventional radiology were not different between groups. Post-operatively, the groups were also similar in regards to the duration of mechanical ventilation and both intensive care unit and for overall transplant hospitalization length of stay. The retransplantation rate was not significantly different. There was a trend for a more expensive adjusted cost of transplant admission for patients in the pLT group (KPE $588,887.00 (466,829.20, 902,360.70) vs. pLT $932,675.30 (668,937.90, 1,120,433.90), *p* = 0.098. 

The 1-year patient survival from both listing and after transplantation was 100% for patients who received a pLT, but this was not significantly different from patients who were transplanted after KPE, although two patients in the KPE group died while on the waitlist. There was no statistical difference in 1-year and 3-year graft survival. 

### 3.3. Does the Era Make A Difference? Pre-Transplant Comparison of Patients Transplanted after KPE or pLT Based on Early Versus Modern Era

In total, 95 of the 97 transplants were done by a single surgeon (R.S.) and a liver transplant operating room team including liver transplant nurses and anesthesiologists. By era, all 51 in the early and 44/46 in the later era were done by the same single surgeon. 

When comparing patients who received a pLT to those who were transplanted after KPE, patients from the KPE group remained significantly younger in both eras at the time of BA diagnosis (early era: KPE 67.9 ± 23.3 vs. pLT 139.4 ± 71.7 days, *p* < 0.001; modern era: KPE 66.1 ± 27.9 vs. pLT 127.9 ± 37.2 days, *p* < 0.001) (Table 3). Although the time between diagnosis and listing was shorter in the pLT group in both eras, it was not significantly different in the early era. However, the difference became significant in the modern era, as patients in the KPE group took longer to be listed than patients in the pLT group (KPE 171.5 ± 110.7 vs. pLT 48.6 ± 40.3 days, *p* = 0.002). The waitlist duration shortened in both groups in the modern era. There was no difference in the number of hospital admissions or days admitted while on the waitlist. Only the number of admissions for cholangitis remained significantly higher in the KPE group. 

### 3.4. Transplant and Post-Transplant Survival Data: Comparison of Patients Transplanted after KPE or pLT Based on Era

The age at transplant improved in the pLT group in the modern era compared to the KPE group, but the difference remained not significant (KPE 326.4 ± 144.2 vs. pLT 261.3 ± 64.6 days, *p* = 0.19) (Table 4). Patients in the pLT group were significantly sicker at the time of transplant as shown by a higher natural PELD score (KPE 16 ± 8 vs. pLT 23 ± 8, *p* = 0.022). The length of the transplant operation shortened in both groups in the modern era, but the operative times, blood loss, ICU and hospital length of stay were quite similar between the two groups. Post-operatively, in the modern era, there was a trend to longer duration of mechanical ventilation for patients in the pLT group (KPE 7.7 ± 6.8 vs. pLT 12.3 ± 9.2 days, *p* = 0.085). The number of readmissions to the intensive care unit were significantly more frequent in the early era in the pLT group (KPE 8/51, 15.7% vs. pLT 3/5, 60%, *p* = *0*.017). While the proportion of ICU readmissions remained higher in the pLT in the modern era, this was no longer significant (KPE 7/46, 15.2 vs. pLT 3/9, 33.3%, *p* = 0.2). Overall, there was no difference in 1-year or 3-year patient and graft survival between groups. 

### 3.5. Comparison of Patients Transplanted after KPE Based on Era 

There was no difference in age at KPE (early 67.9 ± 23.3 vs. modern 66.1 ± 27.9 days, *p* = 0.73) (Table 5). However, the time to listing became significantly longer in the modern era (early 112.1 ± 95.6 vs. modern 171.5 ± 110.7 days, *p* = 0.005). The number of hospital admissions were similar in the two eras as were reasons for admission except for a higher incidence of hospital admissions for malnutrition (including initiation of tube feeds or parental nutrition) in the modern early (early 5/51, 9.8% vs. modern 19/48, 39.6%, *p* = 0.001). Two deaths occurred on the waitlist in the modern era (*p* = 0.14). 

KPE patients were older at transplant in the modern era, although not significantly so (early 297.9 ± 144.0 vs. modern 326.4 ± 144.2 days, *p* = 0.21) (Table 6). The length of transplant surgery shortened by almost an hour in the modern era (early 467.8 ± 89.3 vs. modern 417.4 ± 74.7 min, *p* = 0.005) and blood transfusion requirements diminished, although not significantly (early 162.7 ± 142.1 vs. modern 123.1 ± 92.9 cc/kg, *p* = 0.11). The rate of retransplantation improved significantly (early 8/51, 15.7% vs. modern 1/46, 2.2%, *p* = 0.022), and therefore, the 1-year and 3-year graft survival improved significantly in the modern era. 

### 3.6. Results of pLT: Comparison of Patients Who Received A pLT Based on Era 

There were no significant differences in demographic variables when comparing patients of the pLT based on era (Table 7). Although both the time to listing (early 63.8 ± 69.3 vs. modern 48.6 ± 40.3 days, *p* = 0.61) and the waitlist duration (early 130.0 ± 83.1 vs. pLT 69.0 ± 47.4 days, *p* = 0.1) were shorter, the number of patients was small and did not reach statistical significance. There was a trend in more admissions for malnutrition in the modern era (early 0/5, 0% vs. modern 4/9, 44.4%, *p* = 0.078). 

Although patients in the modern era were transplanted faster, the difference was not significant (age at transplant early 333.2 ± 98.6 vs. modern 261.3 ± 64.6 days, *p* = 0.12) (Table 8). Length of surgery and transfusion requirements improved with time, but also not significantly. Ventilation days, ICU stay and hospital stay were shorter in the early era, but this is due to one early patient being excluded from those analyses, as he was chronically ventilated through a tracheostomy and remained in the ICU until his discharge from the hospital. The rate of retransplantation improved significantly in the modern era (early 2/5, 40.0% vs. 0/9, 0%, *p* = 0.04). The patient and graft survival were similar between eras. 

## 4. Discussion

Primary liver transplant for children with biliary atresia is usually reserved for those children who present with advanced liver disease at the time of diagnosis. Reported incidences of pLT vary between 3–16% [9,13,14,15,16,17,18], but have been as low as less than 1% in Japan [19,20] and as high as 40% in Brazil [11]. Comparing these incidences is challenging given the different rates of organ donation and organ availability in different cultures and countries. Additionally, some studies compare pLT to all patients transplanted after KPE regardless of the timing of when a patient is ultimately listed for a transplant. Additionally, the denominator for these studies varies: some use the total number of patients with BA managed at their institution (regardless of their management), while others only include patients with BA who were ultimately transplanted. In our institutional experience spanning over 23 years, the overall rate of pLT was 10.7%, comparable to previously published data. 

Our study is purposefully limited to the comparison of outcomes in children after pLT to those children who have had unsuccessful KPE and have had to undergo liver transplant within two years of a failed KPE. It does not include comparisons to older children who have had a successful KPE, since we wanted to focus on a population of children who have derived no ostensible benefit from the KPE, and who, with the appropriate, albeit yet unknown selection criteria, might have been spared an unnecessary surgery.

While some studies have reported that transplants after KPE are more complex (higher blood transfusion volumes, longer operative time and increased rate of bowel perforations due to the post-operative adhesions), the differences were actually not statistically significant [12,18,21,22]. Our study showed comparable results between transplant after KPE or pLT. This was felt to be related to the increased surgical experience in transplanting patients after KPE and not to any significant paradigm shifts in the post-operative management either after the KPE or the transplant. The surgical team remained constant over the time span of both eras examined.

Another factor testifying to the increased surgical experience in transplanting complex patients is the statistically lower rate of retransplantation in the modern era in both the KPE and pLT groups. 

Patients in the pLT group had a higher PELD score in the modern era and presented at a later age in both eras. Despite these disadvantages, results in the pLT group were comparable to the KPE group. This may explain the trend in a higher transplant admission cost for the pLT group. The authors recognize that the PELD score is an imperfect metric to reflect the severity of disease in children that underestimates pediatric waitlist mortality [23] and that modifications to the scoring are needed to better attest of the status of patients, and potentially decrease the request of exception points. However, one might speculate that if pLT were done in more patients who presented earlier and would normally be considered for a KPE, the overall morbidity would be reduced below what was observed in both groups in the present study. The key is how to select those 30–50% of patients who fail the KPE within two years so they could be spared surgery with no apparent benefit.

A higher PELD score in patients who received a pLT in the modern era did not translate into a higher number of admissions while on the liver transplant waitlist. Conversely, having undergone a previous KPE did not affect the rate of hospital admissions for complications of ESLD except for admissions for cholangitis. However, while all patients in the pLT group were diagnosed and managed at our institution until transplant, 45% of patients in the KPE group had their KPE done at an outside institution. Therefore, hospital admissions that occurred at outside institutions while being active on the transplant waitlist were not captured in our analysis and could explain why the difference in cost of admissions was not significantly lower for the pLT group. 

Improvements in both post-transplant medical as well as surgical care has led to excellent survival after liver transplant for pediatric patients with BA [24]. Our experience with pLT showed an excellent 100% patient survival at 1 year from listing and 1 year after transplant, and comparable 3-year patient and graft survival to patients transplanted after KPE. This was similar to the findings from other published studies [9,25,26]. However, a recent study reported superior long-term survival outcomes for pLT [10]. This study was done by using a large database and based its patient selection on billing codes for diagnosis and surgical procedures. Surprisingly, their rate of pLT was 50% which is much higher than any previously reported rates and calls into question the accuracy of the methods. Additionally, it does not take into account patients who underwent their KPE in another state [27]. While single-center retrospective studies lack the power to show significant association, large database studies lack granularity, accuracy and stringent data verification processes, and their results should, therefore, be interpreted with caution. 

It is currently difficult, if not impossible, to predict which patient with BA will experience early failure after KPE. The development of a predictive score based on pre-KPE factors would help identify patients without ESLD at high risk for early failure in whom a pLT could be recommended. Pre-KPE histological criteria have been proposed as a means to predict successful bile drainage after a KPE [28], but it has been difficult to reproduce those results. A Taiwanese study suggested a pLT be discussed with parents of children with BA unless they have no living donor available [26]. Suggesting a higher number of patients may undergo a pLT would raise the question of organ shortage and worsening waitlist mortality. Additionally, the waitlist mortality is already the highest in patients less than 1 year of age [29]. In our study, the only waitlist mortalities occurred in children listed after KPE. The authors believe that if policies were established to ensure the splitting of all liver suitable for split liver transplant (intent-to-split policy), the waitlist mortality could be significantly reduced, as has been shown in other countries [30], despite potentially increasing the number of pLT. Segmental grafts have been shown to have beneficial post-transplant outcomes, including reduced incidence of hepatic artery thrombosis due to the large size of donor vessels [31]. ABO incompatible liver transplantation can also be used safely in infants less than 12 months given the immaturity of their immune system [32]. Lastly, promoting living donation in centers able to perform technical variant graft transplants would help reduce the waitlist mortality. 

The Kasai operation remains the treatment of choice at the moment for all babies diagnosed with biliary atresia unless the child demonstrates clear signs of deteriorating liver function. However, in studies from many centers, it has been demonstrated that there is a high failure rate of the KPE even in children with early diagnosis and before the onset of liver failure [7]. Even though our results show no obvious disadvantage in doing the actual transplant operation after a failed KPE, those children will have been subjected to a prior operation that yielded no benefit, with considerable expenditures and with the obvious consequences of suffering through a major operation. The key to adopting a more selective use of the KPE in the treatment of children with BA is to develop accurate and reliable predictors of failure in the approximately 30% of children who need a transplant within 2 years of the KPE. Until the success of the KPE in delaying the need for a transplant approaches 100% success either by immunological or anti-proliferative adjuncts to surgery, a primary liver transplant with a success rate that approaches 100% should be considered in any child who would be predicted to have an unfavorable result after a KPE.

The authors recognize limitations to their study. It is a small retrospective single center study. However, as mentioned earlier, while it lacks the power of a large population sample, it allows for thorough data verification and accuracy when compared to large databases results. The study period extended over 23 years and the management of patients with BA has evolved over time. However, it was not as different as the modern management as other studies who reported and compared the use of other biliary drainage procedure than KPE. 

In conclusion, primary liver transplantation leads to similar outcomes when compared to transplant after early failure of a Kasai portoenterostomy with less mortality on the waitlist. It is possible that a larger multicenter retrospective review followed by a prospective study may show the benefits of performing a primary liver transplant in selected children who are predicted to have a poor outcome after a Kasai procedure. 

## Figures and Tables

**Figure 1 jcm-11-03012-f001:**
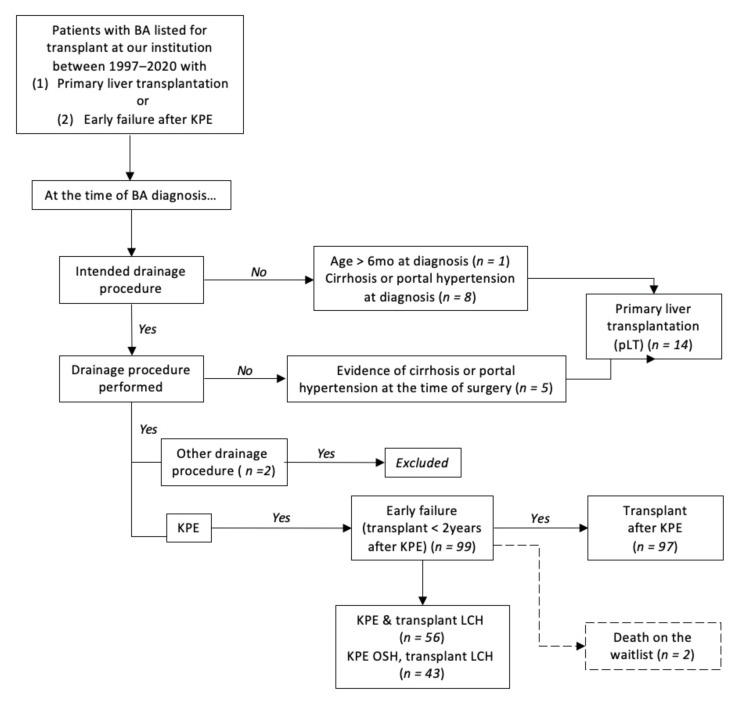
Flowchart describing patient selection (legend: BA: biliary atresia; KPE: Kasai portoenterostomy; LCH: Lurie Children’s Hospital; OSH: Outside hospital; pLT: Primary liver transplantation).

**Table 1 jcm-11-03012-t001:** Pre-transplant comparison of patients transplanted after KPE or as pLT: (*) represents a statistically significant result; (‡) the pre-transplant admission cost comparison only included patients managed since 2009 who received their KPE at our institution (n = 31) and patients from the pLT group (*n* = 10).

Variables	KPE (*n* = 99)	pLT (*n* = 14)	*p* Value
Age at KPE or diagnosis BA (days) (mean ± sd)	67.1 ± 25.5	141.5 ± 46.0	<0.001 *
Time to listing (days) (mean ± sd)	140.8 ± 102.8	44.5 ± 44.7	0.001 *
Waitlist time (days) (mean ± sd)	105.6 ± 102.8	90.8 ± 66.5	0.6
Hospital admissions while on waitlist (*n*, mean ± sd)	3.2 ± 3.3	2.6 ± 2.4	0.58
Days admitted while on waitlist (mean ± sd)	24.9 ± 31.2	22.4 ± 17.6	0.78
Adjusted cost of hospital admissions on the waitlist ‡ (median (IQR))	425,090.00 (285,282.83, 566,405.40)	253,004.10 (95,640.49, 431,530.70)	0.07
Admission on the waitlist (*n*, %): Cholangitis	31 (31.3)	0	0.014 *
Admission on the waitlist (*n*, %): Infections (other than cholangitis)	41 (41.4)	6 (42.9)	0.92
Admission on the waitlist (*n*, %): Gastrointestinal bleeding	20 (20.2)	2 (14.3)	0.6
Admission on the waitlist (*n*, %): Ascites or Spontaneous bacterial peritonitis	25 (25.3)	5 (35.7)	0.41
Admission on the waitlist (*n*, %): Malnutrition	24 (24.2)	4 (28.6)	0.73
Death on the waitlist (yes) (*n*, %)	2 (2.0)	0	0.59

**Table 2 jcm-11-03012-t002:** Post-transplant comparison of patients transplanted after KPE or as pLT. Legend: PELD: Pediatric End-stage Liver Disease; LDLT: Living donor liver transplantation; ICU: intensive care unit.

Variables	KPE (*n* = 97)	pLT (*n* = 14)	*p* Value
Age at transplant (days) (mean ± sd)	311.4 ± 144.0	287.0 ± 82.7	0.54
Calculated PELD score at transplant (mean ± sd)	22 ± 11	27 ± 8	0.15
Type of donor (LDLT) (*n*, %)	39 (40.2)	4 (28.6)	0.4
Transplant surgery duration (minutes) (mean ± sd)	443.9 ± 98.6	423.1 ± 70.0	0.39
Intraoperative pRBC transfusion (cc/kg) (mean ± sd)	143.9 ± 122.3	136.1 ± 137.2	0.83
Duration of mechanical ventilation (days) (mean ± sd)	7.5 ± 6.6	9.8 ± 9.1	0.27
Length of ICU stay (days) (mean ± sd)	14.2 ± 25.6	14.2 ± 10.5	1
Length of hospital stay (days) (mean ± sd)	31.8 ± 36.0	32.8 ± 15.2	0.92
Return to ICU after transplant (*n*, %)	15 (15.5)	5 (35.7)	0.065
Surgical take back post-transplant (mean ± sd)	0.7 ± 1.1	1.0 ± 1.0	0.31
Adjusted cost of transplant admission (median (IQR))	588,887.00 (466,829.20, 902,360.70)	932,675.30 (668,937.90, 1,120,433.90)	0.098
Retransplant (yes) (*n*, %)	8 (8.2)	2 (14.3)	0.46
1-year patient survival from list date (*n*, %)	93 (93.9)	14 (100.0)	0.35
1-year post-transplant patient survival (*n*, %)	91 (93.8)	14 (100.0)	0.35
3-year post-transplant patient survival (*n*, %)	88 (90.7)	13 (92.9)	0.8
1-year post-transplant graft survival (*n*, %)	88 (90.7)	13 (92.9)	0.78
3-year post-transplant graft survival (*n*, %)	86 (88.7)	12 (85.7)	0.78

**Table 3 jcm-11-03012-t003:** Pre-transplant comparison of patients transplanted after KPE or as pLT based on era: demographics, diagnosis of BA and waitlist-related data. Legend: (*) represents a statistically significant result.

	Early Era (1997–2008)			Modern Era (2009–2020)		
Variables	KPE (*n* = 51)	pLT (*n* = 5)	*p* value	KPE (*n* = 48)	pLT (*n* = 9)	*p* Value
Age at KPE or diagnosis BA (days) (mean ± sd)	67.9 ± 23.3	139.4 ± 71.7	<0.001 *	66.1 ± 27.9	127.9 ± 37.2	<0.001 *
Time to listing (days) (mean ± sd)	112.1 ± 95.6	63.8 ± 69.3	0.28	171.5 ± 110.7	48.6 ± 40.3	0.002 *
Waitlist time (days) (mean ± sd)	117.8 ± 121.9	130.0 ± 83.1	0.83	92.6 ± 76.7	69.0 ± 47.4	0.38
Number of hospital admissions while on waitlist (*n*, mean ± sd)	2.8 ± 3.3	2.0 ± 2.0	0.61	3.6 ± 3.4	3 ± 2.4	0.62
Number of days admitted while on waitlist (mean ± sd)	23.1 ± 34.0	22.8 ± 23.9	0.99	26.1 ± 28.3	19.7 ± 16.0	0.53
Admission on the waitlist (*n*, %): Cholangitis	15 (29.4)	0	0.16	16 (33.3)	0	0.041 *
Admission on the waitlist (*n*, %): Infections (other than cholangitis)	20 (39.2)	3 (60.0)	0.37	21 (43.8)	3 (33.3)	0.56
Admission on the waitlist (*n*, %): Gastrointestinal bleeding	11 (21.6)	0 (0)	0.25	9 (18.8)	3 (33.3)	0.81
Admission on the waitlist (*n*, %): Ascites or Spontaneous bacterial peritonitis	10 (19.6)	1 (20.0)	0.98	14 (29.2)	4 (44.4)	0.37
Admission on the waitlist (*n*, %): Malnutrition	5 (9.8)	0 (0)	0.46	19 (39.6)	4 (44.4)	0.79
Death on the waitlist (*n*, %)	0 (0)	0	—	2 (4.2)	0	0.53

**Table 4 jcm-11-03012-t004:** Post-transplant comparison of patients transplanted after KPE or as pLT based on era: transplant-related data and post-transplant outcomes. Legend: (*) represents a statistically significant result.

	Early Era (1997–2008)			Modern Era (2009–2020)		
Variables	KPE (*n* = 51)	pLT (*n* = 5)	*p* Value	KPE (*n* = 46)	pLT (*n* = 9)	*p* Value
Age at transplant (days) (mean ± sd)	297.9 ± 144.0	333.2 ± 98.6	0.6	326.4 ± 144.2	261.3 ± 64.6	0.19
Calculated PELD score at transplant (mean ± sd)	17 ± 9	18 ± 0	0.78	16 ± 8	23 ± 8	0.022 *
Type of donor (LDLT) (*n*, %)	25 (49.0)	1 (20.0)	0.21	14 (30.4)	3 (33.3)	0.86
OR duration (minutes) (mean ± sd)	467.8 ± 89.3	451.2 ± 76.9	0.69	417.4 ± 74.7	407.4 ± 64.3	0.71
Intraoperative pRBC transfusion (cc/kg) (mean ± sd)	162.7 ± 142.1	156.7 ± 176.4	0.93	123.1 ± 92.9	124.7 ± 120.9	0.96
Days on the ventilator (days) (mean ± sd)	7.3 ± 6.4	2.0 ± 2.0	0.64	7.7 ± 6.8	12.3 ± 9.2	0.085
Length of ICU stay (days) (mean ± sd)	15.0 ± 34.1	7.0 ± 2.3	0.67	13.3 ± 10.2	17.6 ± 11.1	0.27
Length of hospital stay (days) (mean ± sd)	32.9 ± 40.1	24.3 ± 18.1	0.1	30.7 ± 31.5	34.7 ± 13.9	0.32
Return to ICU after transplant (*n*, %)	8 (15.7)	3 (60.0)	0.017 *	7 (15.2)	3 (33.3)	0.2
Retransplant (*n*, %)	8 (15.7)	2 (40.0)	0.18	1 (2.2)	0 (0)	0.66
1-year patient survival from list date (*n*, %)	48 (94.1)	5 (100.0)	0.58	45 (93.8)	9 (100.0)	0.45
1-year post-transplant patient survival (*n*, %)	46 (90.2)	5 (100.0)	0.48	45 (97.8)	9 (100.0)	0.66
3-year post-transplant patient survival (*n*, %)	44 (86.3)	5 (100.0)	0.39	44 (95.7)	8 (88.9)	0.4
1-year post-transplant graft survival (*n*, %)	43 (84.3)	4 (80.0)	0.86	45 (97.8)	9 (100.0)	0.66
3-year post-transplant graft survival (*n*, %)	42 (82.4)	4 (80.0)	0.94	44 (95.7)	8 (88.9)	0.4

**Table 5 jcm-11-03012-t005:** Pre-transplant comparison of patients transplanted after KPE based on era: demographics, diagnosis of BA and waitlist-related data. Legend: (*) represents a statistically significant result.

	KPE		
Variables	Early Era (1997–2008) (*n* = 51)	Modern Era (2009–2020) (*n* = 48)	*p* Value
Age at KPE (days) (mean ± sd)	67.9 ± 23.3	66.1 ± 27.9	0.73
Time to listing (days) (mean ± sd)	112.1 ± 95.6	171.5 ± 110.7	0.005 *
Waitlist time (days) (mean ± sd)	117.8 ± 121.9	92.6 ± 76.7	0.22
Number of hospital admissions while on waitlist (mean ± sd)	2.7 ± 3.3	2.0 ± 2.0	0.17
Number of days admitted while on waitlist (mean ± sd)	23.1 ± 34.0	26.1 ± 28.3	0.64
Admission on the waitlist (*n*, %): Cholangitis	15 (29.4)	16 (33.3)	0.67
Admission on the waitlist (*n*, %): Infections (other than cholangitis)	20 (39.2)	21 (43.8)	0.65
Admission on the waitlist (*n*, %): Gastrointestinal bleeding	11 (21.6)	9 (18.8)	0.73
Admission on the waitlist (*n*, %): Ascites or Spontaneous bacterial peritonitis	10 (19.6)	14 (29.2)	0.27
Admission on the waitlist (*n*, %): Malnutrition	5 (9.8)	19 (39.6)	0.001 *
Death on the waitlist (*n*, %)	0 (0)	2 (4.2)	0.14

**Table 6 jcm-11-03012-t006:** Post-transplant comparison of patients transplanted after KPE based on era: transplant-related data and post-transplant outcomes. Legend: * represents a statistically significant result.

	KPE		
Variables	Early Era (1997–2008) (*n* = 51)	Modern Era (2009–2020) (*n* = 46)	*p* Value
Age at transplant (days) (mean ± sd)	297.9 ± 144.0	326.4 ± 144.2	0.21
Calculated PELD score at transplant (mean ± sd)	17 ± 9	16 ± 8	0.46
Type of donor (LDLT) (*n*, %)	25 (49.0)	14 (30.4)	0.06
OR duration (minutes) (mean ± sd)	467.8 ± 89.3	417.4 ± 74.7	0.005 *
Intraoperative pRBC transfusion (cc/kg) (mean ± sd)	162.7 ± 142.1	123.1 ± 92.9	0.11
Days on the ventilator (days) (mean ± sd)	7.3 ± 6.4	7.7 ± 6.8	0.84
Length of ICU stay (days) (mean ± sd)	15.0 ± 34.1	13.3 ± 10.2	0.73
Length of hospital stay (days) (mean ± sd)	32.9 ± 40.1	30.7 ± 31.5	0.73
Return to ICU after transplant (*n*, %)	8 (15.7)	7 (15.2)	0.95
Retransplant (*n*, %)	8 (15.7)	1 (2.2)	0.022 *
1-year patient survival from list date (*n*, %)	48 (94.1)	45 (93.8)	0.94
1-year post-transplant patient survival (*n*, %)	46 (90.2)	45 (97.8)	0.13
3-year post-transplant patient survival (*n*, %)	44 (86.3)	44 (95.7)	0.13
1-year graft survival (*n*, %)	43 (84.3)	45 (97.8)	0.025 *
3-year graft survival (*n*, %)	42 (82.4)	44 (95.7)	0.044 *

**Table 7 jcm-11-03012-t007:** Pre-transplant comparison of patients transplanted after pLT based on era: demographics, diagnosis of BA and waitlist-related data.

	pLT		
Variables	Early Era (1997–2008) (*n* = 5)	Modern Era (2009–2020) (*n* = 9)	*p* Value
Age at diagnosis BA (days) (mean ± sd)	139.4 ± 71.7	127.9 ± 37.2	0.70
Time to listing (days) (mean ± sd)	63.8 ± 69.3	48.6 ± 40.3	0.61
Waitlist time (days) (mean ± sd)	130.0 ± 83.1	69.0 ± 47.4	0.10
Number of hospital admissions while on waitlist (mean ± sd)	3.6 ± 3.4	3 ± 2.4	0.45
Number of days admitted while on waitlist (mean ± sd)	22.8 ± 23.9	19.8 ± 16.0	0.79
Admission on the waitlist (*n*, %): Cholangitis	0	0	—
Admission on the waitlist (*n*, %): Infections (other than cholangitis)	3 (60.0)	3 (33.3)	0.33
Admission on the waitlist (*n*, %): Gastrointestinal bleeding	0 (0)	3 (33.3)	0.15
Admission on the waitlist (*n*, %): Ascites or Spontaneous bacterial peritonitis	1 (20.0)	4 (44.4)	0.36
Admission on the waitlist (*n*, %): Malnutrition	0 (0)	4 (44.4)	0.078
Death on the waitlist (*n*, %)	0	0	—

**Table 8 jcm-11-03012-t008:** Post-transplant comparison of patients transplanted after pLT based on era: transplant-related data and post-transplant outcomes (waitlist). Legend: (*) represents a statistically significant result.

	pLT		
Variables	Early Era (1997–2008) (*n* = 5)	Modern Era (2009–2020) (*n* = 9)	*p* Value
Age at transplant (days) (mean ± sd)	333.2 ± 98.6	261.3 ± 64.6	0.12
Calculated PELD score at transplant (mean ± sd)	18 ± 0	23 ± 8	0.18
Type of donor (LDLT) (*n*, %)	1 (20.0)	3 (33.3)	0.60
OR duration (minutes) (mean ± sd)	451.2 ± 76.9	407.4 ± 64.3	0.28
Intraoperative pRBC transfusion (cc/kg) (mean ± sd)	156.7 ± 176.4	124.7 ± 120.8	0.69
Days on the ventilator (days) (mean ± sd)	2.0 ± 2.0	12.3 ± 9.2	0.099
Length of ICU stay (days) (mean ± sd)	7.0 ± 2.3	17.6 ± 11.1	0.12
Length of hospital stay (days) (mean ± sd)	24.3 ± 18.1	34.7 ± 13.9	0.28
Return to ICU after transplant (*n*, %)	3 (60.0)	3 (33.3)	0.33
Retransplant (*n*, %)	2 (40.0)	0 (0)	0.04 *
1-year patient survival from list date (*n*, %)	5 (100.0)	9 (100.0)	–
1-year post-transplant patient survival (*n*, %)	5 (100.0)	9 (100.0)	–
3-year post-transplant patient survival (*n*, %)	5 (100.0)	8 (88.9)	0.40
1-year post-transplant graft survival (*n*, %)	4 (80.0)	9 (100.0)	0.18
3-year post-transplant graft survival (*n*, %)	4 (80.0)	8 (88.9)	0.68

## Data Availability

The data that support the findings of this study are available from the corresponding author upon reasonable request.

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
