# Peer review of "Primary Liver Transplantation vs. Transplant after Kasai Portoenterostomy for Infants with Biliary Atresia"

_jcm, 2022, doi:10.3390/jcm11113012_

Round 1

Reviewer 1 Report

This article introduced the data of children with biliary atresia within 2 years of sequential transplantation and children with primary liver transplantation in the single-center transplantation database. Moreover, they performed subgroup analysis of the same group of patients according to different eras. They conclude that for BA patients with severe liver cirrhosis, primary transplantation had certain advantages.

1 The data collection of the article is selective. Only children who underwent kasai surgery and transplantation within 2 years after the first operation were selected for this article. They did not collect children who underwent kasai surgery and survived in their native liver for more than 2 years. Although primary liver transplantation may be an option, it is difficult to distinguish which patients would benefit more from PLT. If data on children who failed early and survived longer could be compared, there might be some helpful results. Unfortunately, I couldn't find it in the article. The operation time, blood loss, postoperative survival rate and CPI of patients in early failure group and in the primary transplant group were comparable. However, these data can’t lead to the conclusions mentioned in the text.

2 According to the era, the authors divided the treatment into two phases. The operation time was significantly shortened in the later stage, and the operation effect was better. These are only database data, and the detailed reasons, such as whether the two stages are the same group of doctors, and what are the differences in the techniques used, are not mentioned in this article. This makes it much less meaningful to compare them with each other. Therefore, I personally think that this part of the content can be appropriately reduce and transplantation indications and postoperative complications might be discussed.

  1. " A multicenter collaboration may help demonstrate the potential benefits of PLT in patients predicted to have early failure of KPE ". This is just a guess, not the conclusion of the article.
  2. In the Discussion section (paragraph 7), " it recommended abandoning KPE as the first-line treatment for all BA patients due to the near 100% success rate of PLT and 30% failure rate of KPE." However, liver transplantation is not without short-term and long-term complication. How to avoid unnecessary KPE, the article obviously does not elaborate on its operability.

Author Response

  1. The data collection of the article is selective. Only children who underwent Kasai surgery and transplantation within 2 years after the first operation were selected for this article. They did not collect children who underwent Kasai surgery and survived in their native liver for more than 2 years.

The reviewer points out that children who survived longer than 2 years were not included in this study. The reason for this is that those who survived longer obviously benefitted from the portoenterostomy and therefore did not need a transplant within the first two years of life.The two-year period as a cut-off was chosen because children with no bile drainage after their porto-enterostomies would generally be expected to die within two years of the operation, and thus would be comparable in survival to those with no Kasai operation.

We could definitely compare data with children who survive longer than two years after the Kasai, but we felt that these would be older children, larger in size, and therefore not comparable as a control group to the primary transplant group who were all infants or babies under the age of two years. We set out to compare the patients who presented too late for a Kasai operation to those who derived no benefit from the operation.

The reviewer suggests comparing data on children who fail early but survive longer, but there is no data on children who fail early and survive longer because all early failures are transplanted or die waiting for a transplant in North America.

Although primary liver transplantation may be an option, it is difficult to distinguish which patients would benefit more from PLT. If data on children who failed early and survived longer could be compared, there might be some helpful results. Unfortunately, I couldn't find it in the article.

 No one survived longer after a failed procedure without a liver transplant. Patients were either transplanted or died on the waiting list. The reviewer suggests comparing data on children who fail early but survive longer, but there is no data on children who fail early and survive longer because all early failures are transplanted or die waiting for a transplant in North America.

We have added this to the manuscript.

The operation time, blood loss, postoperative survival rate and CPI of patients in early failure group and in the primary transplant group were comparable. However, these data can’t lead to the conclusions mentioned in the text.

Our conclusion simply states that primary transplantation has similar outcomes to transplant after a failed portoenterostomy, and that conclusion is firmly supported by our single center review. The survival in the primary transplant group was 100% at 1 year after transplant compared to 93.8% in the kasai portoenterostomy group. We can modify the statement regarding what follows the conclusion in regard to the place of KPE and whether it is appropriate in all patients with BA.

  1. According to the era, the authors divided the treatment into two phases. The operation time was significantly shortened in the later stage, and the operation effect was better. These are only database data, and the detailed reasons, such as whether the two stages are the same group of doctors, and what are the differences in the techniques used, are not mentioned in this article. This makes it much less meaningful to compare them with each other. Therefore, I personally think that this part of the content can be appropriately reduce and transplantation indications and postoperative complications might be discussed.

We have amended the methods section to reflect what changes were made between the two eras. The kasai operations were done primarily by a single surgeon over the course of the two eras as were the liver transplants. There were no major paradigm changes over the course of the study period either in the treatment of babies with biliary atresia or in the treatment of children with liver transplants. Liver transplant surgical care did change in terms access to technical variant grafts perhaps accounting for the decrease waiting times seen in both groups, but the benefit affected both groups equally. We have amended the methods section to reflect this.

In regard to the reviewer # 1 comments about comparing the two eras in tables 5-8 where we compare the results and changes in the two eras to each group to itself, we feel it is useful to leave those in since it allows for a more granular look at the changes that accounted for the pre and post-transplant results.

  1. " A multicenter collaboration may help demonstrate the potential benefits of PLT in patients predicted to have early failure of KPE ". This is just a guess, not the conclusion of the article.

We have reworded the conclusion

4.In the Discussion section (paragraph 7), " it recommended abandoning KPE as the first-line treatment for all BA patients due to the near 100% success rate of PLT and 30% failure rate of KPE." However, liver transplantation is not without short-term and long-term complication. How to avoid unnecessary KPE, the article obviously does not elaborate on its operability.

We never recommended abandoning the Kasai operation at any point in the manuscript. We simply recommended a more selective use of the operation given the high 30% failure rate. We have also re-worded the discussion to reflect how this could be done. 

Reviewer 2 Report

This study demonstrated the data comparing between primary liver transplantation and transplantation after Kasai operation and those between eras in patients with biliary atresia. The research topic is a tremendous clinical interest l in this field. Although it is a single institute retrospective nature, the data showed in this article would be informative for physicians who take care of binary atresia.
However, the authors need to see their data more fairly. According to their data, there seemed no benefit for primary liver transplantation compared to transplantation after failed-Kasai. Besides, since there are no apparent, predictable factors for the requirement of transplantation after Kasai, there might have been patients for whom liver transplantation would have been unnecessary among the primary liver transplantation group.

I recommend reconsidering below.

Major 
 The statement in Discussion [Although the statistics in our current study-avoided by a needless prior operation] sounds injudicious argument. Considering that no other field of surgery other than transplantation requires organs from other individuals, Kasai's operation should be justified even if there is a 30% chance to fail,
especially in the situation where Kasai would not add more risks to subsequent liver transplantation.

Even though the authors failed to show the data in this study, and if they still believe that primary liver transplantation rather than Kasai would be more beneficial for patients with biliary atresia; the argument here should be 
1, What advantages in the primary liver transplantation do they believe for what subset in biliary atresia patients.
2, Why they fail to show that.
 As well as
3, Whether operative findings such as cirrhosis and portal hypertension were applicable to select for primary liver transplantation.

Minor
Figure 1.
Regarding the most bottom box showing 
KPE and transplant LCH (n=45)
KPE OSH, transplant LCH (n=44)
What is OSH?
The total number is 89. Is it correct?
The arrow from the box above (Early failure) to this box may be unnecessary. Please consider removing it and describe it in Material and Methods.

Author Response

Major 
 The statement in Discussion [Although the statistics in our current study-avoided by a needless prior operation] sounds injudicious argument. Considering that no other field of surgery other than transplantation requires organs from other individuals, Kasai's operation should be justified even if there is a 30% chance to fail,
especially in the situation where Kasai would not add more risks to subsequent liver transplantation.

We can amend this statement to emphasize that research is necessary to predict which children will have a failure after a Kasai. It is true that in every major series of the results of the KPE operation there are a high (30-40%_ failure rate. We are simply stating that pediatric surgeons should think about how this result can be improved. If we could do that without a liver transplant option, then that would be worth pursuing.

Even though the authors failed to show the data in this study, and if they still believe that primary liver transplantation rather than Kasai would be more beneficial for patients with biliary atresia; the argument here should be 
1, What advantages in the primary liver transplantation do they believe for what subset in biliary atresia patients.
2, Why they fail to show that.
 As well as
3, Whether operative findings such as cirrhosis and portal hypertension were applicable to select for primary liver transplantation.

We will try and explain our discussion point more adequately. If we could predict who would fail the KPE, the advantages of not doing a KPE would be to avoid needless surgery and the cost to the medical system that this would save.

pLT were done only in children with advanced liver disease as explained in the manuscript in a more detailed manner.

Minor
Figure 1.
Regarding the most bottom box showing 
KPE and transplant LCH (n=45)
KPE OSH, transplant LCH (n=44)
What is OSH?
The total number is 89. Is it correct?
The arrow from the box above (Early failure) to this box may be unnecessary. Please consider removing it and describe it in Material and Methods.

We will correct the error in Figure 1. Thank you for pointing it out.

OSH = Outside hospital. we will add to the legend.

Round 2

Reviewer 1 Report

"A multi- center collaboration with more patients may help demonstrate the potential benefits of pLT in patients predicted to have early failure of KPE." is just authors' imagination. I can't find any results in this paper can make this conclusion.

Reviewer 2 Report

This revised manuscript emphasized the necessity of the indicators to determine who should undergo primary liver transplantation rather than Kasai's procedure. The factors associated with failed Kasai have been long clinical interests. I hope this manuscript reminds clinicians that we need to elucidate these predictors for children with biliary atresia.